# The Use of a Hybrid Closed-Loop System for Glycemic Control in Two Pediatric Patients with Type 1 Diabetes Undergoing Minor Surgery

**DOI:** 10.3390/healthcare11040587

**Published:** 2023-02-15

**Authors:** Sebastian Seget, Julia Włodarczyk, Wanda Lutogniewska, Ewa Rusak, Maria Dróżdż, Przemysława Jarosz-Chobot

**Affiliations:** Department of Children’s Diabetology, Medical University of Silesia, 40-752 Katowice, Poland

**Keywords:** type 1 diabetes, advanced closed-loop hybrid systems, minor surgery

## Abstract

Technological progress in the treatment of type 1 diabetes requires doctors to use modern methods of insulin therapy in all areas of medicine that patients may come into contact with, including surgical interventions. The current guidelines indicate the possibility of using continuous subcutaneous insulin infusion in minor surgical procedures, but there are few reported cases of using a hybrid closed-loop system in perioperative insulin therapy. This case presentation focuses on two children with type 1 diabetes who were treated with an advanced hybrid closed-loop (AHCL) system during a minor surgical procedure. In the periprocedural period, the recommended mean glycemia and the time in range were maintained.

## 1. Introduction

The incidence of type 1 diabetes is increasing in the world [1]. Patients are increasingly using modern methods in the treatment of diabetes, such as insulin pumps, including advanced hybrid closed-loop (AHCL) systems [2,3]. Due to the growing population of patients treated with continuous subcutaneous insulin infusion (CSII), it is necessary to consider these technologies during hospitalization and when patients require surgery, taking into account the benefits of such treatment.

Numerous studies show the benefits of using AHCL systems in patients’ everyday lives, such as improvement in glycemic control, improved glucose monitoring satisfaction, and life quality [4,5,6,7]. However, there are few reported cases of using hybrid closed-loop (HCL) systems in perioperative insulin therapy. In each of these cases, the use of a hybrid closed-loop system was both safe and effective [8,9]. 

Surgery and anesthesia stress the body in its response to the associated tissue damage. The release of stress hormones leads to fluctuations in glycemia [10]. Patients suffering from type 1 diabetes are especially prone to this problem. Moreover, being in a hospital is usually a stressful experience for children.

Taking into account the above issues, it is necessary to select a method of insulin therapy that ensures the best possible glycemic control and is optimal from the patient’s point of view. 

The current guidelines allow the use of continuous subcutaneous insulin infusion (CSII) during short surgical procedures (lasting < 2 h) [11,12]. Modern insulin pumps enable the use of a continuous glucose monitoring (CGM) system, which allows for more-accurate monitoring of glycemic variability during the procedure [11]. An AHCL system can maintain glycemia at a target level based on the information obtained through continuous glucose monitoring (CGM), using an algorithm to automatically estimate the required insulin dose, along with the manual initiation of post-meal boluses. It should be noted that in the case of a hybrid insulin pump in an AHCL mode, it is possible to change the glycemic targets, including setting the so-called temporary goal [13]. 

Before each surgical procedure, it is necessary to prepare patients as per the current guidelines. Personal insulin pump therapy is allowed although contingent on approval and ability to conduct it by an anesthesiologist. As insulin sensitivity fluctuates, it is recommended to maintain the basal flow appropriate for the time of day and, in the event of hypoglycemia in the patient, to suspend the insulin supply for 30 min or provide a correction bolus in the case of hyperglycemia. The recommended glycemia during the procedure is 100–180 mg/dL [11,12,14].

The aim of the following case studies is to report on the use of an AHCL insulin pump in a girl and a boy with type 1 diabetes during sedoanalgesia with gastrointestinal endoscopy and umbilical hernia repair.

### 1.1. Case 1 

In January 2022, a 10.5-year-old female patient, with type 1 diabetes of a duration of 6 years, was admitted to the Children’s Diabetology Department of GCZD in Katowice. From the moment of diagnosis, she had been treated with an insulin pump MiniMed 640 G (with predictive low-glucose suspend (PLGS) function), and from February 2021, she was treated with the AHCL system (insulin aspart). Her glucose metrics met the criteria for proper glycemic control from the onset of the disease, with HbA1c ranging between 5.8 and 6.4%, and TIR > 80% [11]. The purpose of the hospitalization was to conduct elective diagnostic endoscopy of the gastrointestinal tract for celiac disease with the use of sedoanalgesia. 

The metrics of glycemic control and the insulin therapy performed before the procedure are presented in Table 1.

On 12 January 2022, between 8:00 and 8:20 a.m., gastrointestinal endoscopy was performed. In the periprocedural period, the CSII therapy was continued, with the auto-mode system turned on and a temporary target of 150 mg/dL set; the system was activated at 3:00 a.m. This target was set for 7 h. After that, the target was reverted to 100 mg/dL. The patient ate her last solid meal around midnight on the previous day, and she ate the first meal after the procedure around 11:00 a.m. on 12 January 2022. Hydration without glucose infusion was also used: 500 mL of multi-electrolyte fluid was administered on the day of the procedure from 6:00 a.m. In the periprocedural period, glycemia remained within the desired range (Table 1). The girl was discharged home on the same day the surgery was performed and was in good general condition. 

### 1.2. Case 2 

In May 2022, a 4-year-old boy, diagnosed with type 1 diabetes in 2020, was admitted to the Department of Pediatric Surgery for elective umbilical hernia surgery, which was diagnosed in August 2021. Additionally, he was diagnosed with familial hypercholesterolaemia (2021). Since the diagnosis of type 1 diabetes, he had been treated with an insulin pump MiniMed 640 G (with LPGS function), and from February 2022, he was treated with a MiniMed 780 G system (Fiasp insulin). The most recent HbA1c was 6.9% (March 2022). 

The values of the glycemic control parameters and the pre-treatment insulin therapy are presented in Table 2.

On 26 May 2022, between 8:30 and 9:10 a.m., surgical repair of the umbilical hernia was performed, while insulin therapy was continued using a personal insulin pump with an AHCL function. In the periprocedural period, the CSII therapy was continued, with the auto-mode system being set with a temporary target of 150 mg/dL and activated at 3:00 a.m. This target was set for 7 h after activation. Afterward, the target was reverted to 110 mg/dL. The patient ate his last solid meal around 09:00 p.m. on the previous day, and he ate the first meal after the procedure around 11:00 a.m. Hydration without glucose infusion was also used—250 mL of multi-electrolyte fluid was administered on the day of the procedure from 8:00 a.m. During the periprocedural period, glycemia remained within the desired range.

## 2. Discussion

Type 1 diabetes is one of the most common chronic diseases in childhood, and its incidence is increasing [15]. As a result, the number of diabetic patients requiring surgical intervention is also increasing. The technology used in the treatment of type 1 diabetes is constantly advancing, and it is necessary to adapt these systems to various aspects of therapy, including perioperative management. The use of an AHCL system for this purpose is less invasive compared to intravenous insulin infusion and avoids interrupting patients’ routine insulin therapy, while maintaining good glycemic control. However, considering that only fast or short-acting insulin can be used in a pump, it is important to remember that an interruption of the infusion may quickly result in hyperglycemia. Therefore, it is necessary to carefully check if the pump is working properly, that there is no tubing occlusion, and that the settings are correct. When choosing CSII as a method of perioperative insulin therapy, it is highly recommended to consult with an anesthesiologist who will participate in the procedure. The lack of experience with such methods may prevent the use of such management. 

Hyperglycemia during surgery can result from increased production of catecholamines and cortisol, which reduce insulin sensitivity. In addition, drugs used during anesthesia can lead to hyperglycemia. The use of an AHCL system during surgery may allow glucose levels to be maintained at the desired target [16].

In the process of hospitalization of patients, it is worth applying, if possible, the therapy that has already been used by them, reducing the costs of diagnostics and treatment. This is an additional advantage of continuing insulin therapy with an advanced hybrid closed-loop system during surgery.

### Learning Points

Using an AHCL insulin pump system enabled the safe and correct performance of a minor elective surgical procedure (less than two hours) on two young patients with type 1 diabetes, while maintaining good glycemic control.

## Figures and Tables

**Table 1 healthcare-11-00587-t001:** AHCL report 2 weeks prior to hospital admission and on the day of the procedure.

Parameter		Two Weeks Prior to Hospital Admission	The Day of the Procedure from 00.00 a.m. of 12 January 2022 to 00.00 a.m. of 13 January 2022
SmartGuard mode [% time]		100	100
Manual mode [% time]		0	0
Sensor use [% time]		93	100
Mean glycemia ± SD [mg/dL]		112 ± 32	126 ± 35
GMI [%]		6	6.3
Coefficient of variation CV		29.3	28.57
Daily insulin dose [U/day]		24.6	17.3
Daily Boluses [U/day]		17.8	11.1
Automatic correction [U/day]		0.9	1.2
Automatic base [U/day]		6.8	6.2
Grams of carbohydrates/24 h		212 ± 7.6	146
Active insulin [h]		2.00	2.00
Time in range [%]	>250 mg/dL	0	0
	180–250 mg/dL	4	6
	70–180 mg/dL	92	88
	54–70 mg/dL	3	5
	<54 mg/dL	1	1

SmartGuard mode—automated modes; GMI—glucose management indicator.

**Table 2 healthcare-11-00587-t002:** AHCL report 2 weeks prior to hospital admission and on the day of the procedure.

Parameter		Two Weeks Prior to Hospital Admission	The Day of the Procedure from 00.00 a.m. of 26 May 2022 to 00.00 a.m. of 27 May 2022
SmartGuard Mode [% time]		100	100
Manual mode [% time]		0	0
Sensor use [% time]		94	100
Mean sensor glycemia ± SD [mg/dL]		146 ± 56	153 ± 62
GMI [%]		6.8	7.0
Coefficient of variation CV [%]		38,1	40.52
Daily insulin dose [U/day]		9.0	10.1
Bolus [U/day]		5.7	6.6
Automatic correction [U/day]		1.6	1.9
Automatic dose [U/day]		3.3	3.5
Grams of carbohydrates/24 h		126 ± 33	156
Active insulin [h]		2.00	2.00
Time in range [%]	>250 mg/dL	4	9
	180–250 mg/dL	21	21
	70–180 mg/dL	72	70
	54–70 mg/dL	3	0
	<54 mg/dL	0	0

SmartGuard mode—automated modes; GMI—glucose management indicator.

## Data Availability

The authors exclude this claim because no new data were created.

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
