# Peer review of "The Use of a Hybrid Closed-Loop System for Glycemic Control in Two Pediatric Patients with Type 1 Diabetes Undergoing Minor Surgery"

_healthcare, 2023, doi:10.3390/healthcare11040587_

Round 1
Reviewer 1 Report
The manuscript discusses two case reports on surgical procedures on pediatric patients with type 1 diabetes who leverage closed-loop insulin delivery. The aim is to investigate whether glycemic targets are still met during hospitalization and surgery by continuing the use of these closed-loop systems. The stress due to hospitalization and surgery can indeed impair glycemic regulation. The results achieved on two pediatric patients suggest that closed-loop insulin delivery could indeed be safely used during surgical procedures.
In my opinion, the work needs some major adjustments. The quality of the English should be improved and possibly double-checked by an expert speaker. The overall clarity of the work could be improved by better explaining the metrics employed to assess the results. Also, the introduction should be adjusted to better clarify the importance of the study. In particular, the relevance of maintaining closed-loop insulin therapy during surgical procedures should be explained more clearly.
Hereafter are reported more specific comments.
Introduction:
· What do the authors mean by “personal” insulin pumps? Is it a standard CSII therapy with patients manually adjusting insulin infusion?
· The acronym “HCL” was never defined.
· It could be helpful for the reader if the author introduced CGM and CSII beforehand, maybe at the beginning of the introduction, and explain that closed-loop systems consist of a CGM, a CSII and a closed-loop control algorithm that modulates basal insulin infusion. It was never explained in the manuscript that sensor-augmented pumps (like the Minimed 640G, cited in text) can suspend insulin and AHCL systems can modulate it only because they can exploit CGM measurements.
· The authors stated that basal flow should be adjusted based on the time of the day, but did not explain why. It could be helpful for the reader if the authors explained that insulin sensitivity changes along the day.
Case 1, Case 2:
· The metrics reported in Tables 1 and 2 should be shortly explained beforehand.
· How were the metrics reported under the column “The day of the procedure” computed? Did the authors consider the time interval between 00.00 am of 12/01/22 to 00.00am of 13/01/22 or the time interval between 08:00 am of 12/01/22 to 08.00am of 13/01/22?
· For what I understand, one of the major risks associated with these operations is the incurrence of hypoglycemia due to the release of stress hormones. It would be interesting if the authors provided the data about glycemic levels and hypoglycemia occurrence during the time windows of the operations.
Learning points:
· Since only two patients are considered, I believe the first learning point may be an overstatement.
· For what it concerns the second learning point, I do not understand what part of the study highlights that stress associated to surgery and hospitalization was reduced. Hypoglycemia did not increase, but this is not enough to assume a causal relationship between these two phenomena.
Typos:
· Abstract: extra space between “(AHCL)” and “during”.
· “lead” -> “leads” in line 13, page 1
· A dot is missing in line 9, page 2, after “the procedure (11)”
· A space is missing in line 2, page 3, after “reverted to”.
· “managment” -> “management” in line 12, page 4
Author Response
Thank you very much for Your valuable and accurate observations and comments on the article, which we found very helpful in improving our paper.
The article has been revised and changed accordingly. Please find the specific answers below.
On behalf of all authors,
With kind regards,
Sebastian Seget
The manuscript discusses two case reports on surgical procedures on pediatric patients with type 1 diabetes who leverage closed-loop insulin delivery. The aim is to investigate whether glycemic targets are still met during hospitalization and surgery by continuing the use of these closed-loop systems. The stress due to hospitalization and surgery can indeed impair glycemic regulation. The results
achieved on two pediatric patients suggest that closed-loop insulin delivery could indeed be safely used during surgical procedures. In my opinion, the work needs some major adjustments. The quality of the English should be improved and possibly double-checked by an expert speaker. The overall clarity of the work could be improved by better explaining the metrics employed to assess the results. Also, the introduction should be adjusted to better clarify the importance of the study. In particular, the relevance of maintaining closed-loop insulin therapy during surgical procedures should be explained more clearly.
Thank you very much for your comments. Changes have been made to the text. The article was checked by an expert English speaker.
What do the authors mean by “personal” insulin pumps? Is it a standard CSII therapy with patients manually adjusting insulin infusion?
The change was made in the text.
· The acronym “HCL” was never defined.
The acronym "HCL" was explained
· It could be helpful for the reader if the author introduced CGM and CSII beforehand, maybe at the beginning of the introduction, and explain that closed-loop systems consist of a CGM, a CSII and a closed-loop control algorithm that modulates basal insulin infusion. It was never explained in the manuscript that sensor-augmented pumps (like the Minimed 640G, cited in text) can suspend insulin and AHCL systems can modulate it only because they can exploit CGM measurements.
The change was made in the text.
· The authors stated that basal flow should be adjusted based on the time of the day, but did not explain why. It could be helpful for the reader if the authors explained that insulin sensitivity changes along the day.
The change was made in the text.
Case 1, Case 2:
· The metrics reported in Tables 1 and 2 should be shortly explained beforehand.
· How were the metrics reported under the column “The day of the procedure” computed? Did the authors consider the time interval between 00.00 am of 12/01/22 to 00.00am of 13/01/22 or the time interval between 08:00 am of 12/01/22 to 08.00am of 13/01/22?
The change was made in the text.
· For what I understand, one of the major risks associated with these operations is the incurrence of hypoglycemia due to the release of stress hormones. It would be interesting if the authors provided the data about glycemic levels and hypoglycemia occurrence during the time windows of the operations.
The change was made in the text.
Learning points:
· Since only two patients are considered, I believe the first learning point may be an overstatement.
The change was made in the text.
· For what it concerns the second learning point, I do not understand what part of the study highlights that stress associated to surgery and hospitalization was reduced. Hypoglycemia did not increase, but this is not enough to assume a causal relationship between these two phenomena.
The second learning point has been removed.
Typos:
· Abstract: extra space between “(AHCL)” and “during”.
· “lead” -> “leads” in line 13, page 1
· A dot is missing in line 9, page 2, after “the procedure (11)”
· A space is missing in line 2, page 3, after “reverted to”.
· “managment” -> “management” in line 12, page 4
The change was made in the text.
Reviewer 2 Report
This brief report of two children with well-controlled type 1 diabetes who required minor, brief procedures performed under sedoanalgesia showed that in collaboration with the anesthesiologist, the diabetologist adapted the pump settings and the children continued safely and effectively to use automated insulin delivery via their advanced hybrid closed loop systems.
Tables 1 and 2 show data for the 2 weeks before the procedure and the day of the procedure. I would be interested in knowing the glucose metrics and insulin delivery before, during and after the procedure.
P. 4/5 The last paragraph describes the enormous cost of healthcare in the US, specifically related to diabetes. In my judgment, this is irrelevant to the aim of these case reports.
From my own clinical experience, I agree with the authors' comment that anesthesiologists' lack of experience with AHCL systems is a major barrier to their use. This is understandable because the anesthesiologist is responsible for the patient's safety during the procedure. To overcome this barrier requires close collaboration between the diabetologist (the expert on how the system works) and the anesthesiologist.
The authors did not measure patient stress. How do they know continued use of AHCL during the procedure reduced patient stress?
Author Response
Thank you very much for Your valuable and accurate observations and comments on the article, which we found very helpful in improving our paper.
The article has been revised and changed accordingly. Please find the specific answers below.
On behalf of all authors,
With kind regards,
Sebastian Seget
Tables 1 and 2 show data for the 2 weeks before the procedure and the day of the procedure. I would be interested in knowing the glucose metrics and insulin delivery before, during and after the procedure.
Unfortunately, only such data was obtained from Carelink Professional
P. 4/5 The last paragraph describes the enormous cost of healthcare in the US, specifically related to diabetes. In my judgment, this is irrelevant to the aim of these case reports.
The sentence has been deleted.
From my own clinical experience, I agree with the authors' comment that anesthesiologists' lack of experience with AHCL systems is a major barrier to their use. This is understandable because the anesthesiologist is responsible for the patient's safety during the procedure. To overcome this barrier requires close collaboration between the diabetologist (the expert on how the system works)
and the anesthesiologist.
Thank you for confirming our comments.
The authors did not measure patient stress. How do they know continued use of AHCL during the procedure reduced patient stress?
The sentence has been deleted.